# Preparation, Characterization and In Vitro Stability of a Novel ACE-Inhibitory Peptide from Soybean Protein

**DOI:** 10.3390/foods11172667

**Published:** 2022-09-01

**Authors:** Sara Sangiorgio, Nikolina Vidović, Giovanna Boschin, Gilda Aiello, Patrizia Arcidiaco, Anna Arnoldi, Carlo F. Morelli, Marco Rabuffetti, Teresa Recca, Letizia Scarabattoli, Daniela Ubiali, Giovanna Speranza

**Affiliations:** 1Department of Chemistry, University of Milan, Golgi 19, 20133 Milan, Italy; 2Department of Pharmaceutical Sciences, University of Milan, L. Mangiagalli 25, 20133 Milan, Italy; 3Department of Human Science and Quality of Life Promotion, Telematic University San Raffaele, 00166 Rome, Italy; 4Centro Grandi Strumenti, University of Pavia, Bassi 21, 27100 Pavia, Italy; 5Department of Drug Sciences, University of Pavia, Viale Taramelli 12, 27100 Pavia, Italy

**Keywords:** soybean protein hydrolysate, ACE inhibitor, peptide synthesis, Lineweaver–Burk plot, simulated gastrointestinal digestion, functional food

## Abstract

A soy protein isolate was hydrolyzed with Alcalase^®^, Flavourzyme^®^ and their combination, and the resulting hydrolysates (A, F and A + F) were ultrafiltered and analyzed through SDS-PAGE. Fractions with MW < 1 kDa were investigated for their ACE-inhibitory activity, and the most active one (A < 1 kDa) was purified by semi-preparative RP-HPLC, affording three further subfractions. NMR analysis and Edman degradation of the most active subfraction (A1) enabled the identification of four putative sequences (ALKPDNR, VVPD, NDRP and NDTP), which were prepared by solid-phase synthesis. The comparison of their ACE-inhibitory activities suggested that the novel peptide NDRP might be the main agent responsible for A1 fraction ACE inhibition (ACE inhibition = 87.75 ± 0.61%; IC_50_ = 148.28 ± 9.83 μg mL^−1^). NDRP acts as a non-competitive inhibitor and is stable towards gastrointestinal simulated digestion. The Multiple Reaction Monitoring (MRM) analysis confirmed the presence of NDRP in A < 1 kDa.

## 1. Introduction

Bioactive peptides, i.e., peptides that exert some health benefits beyond their nutritional value, may be already present in foods as natural components, but more frequently are in a latent state within the sequence of the parent protein, from which they can be released during digestion or food processing. These peptides encrypted in the protein sequence are referred to as cryptides [1]. In this regard, the nutritional and functional value of a protein depends on its amino acid content, but also on the presence of cryptides in its sequence.

Enzymatic hydrolysis based on the use of proteases is the most common procedure used to produce bioactive food peptides [2]. The use of proteolytic enzymes is preferred to chemical hydrolysis because reproducible molecular weight profiles and peptide composition are usually obtained. Moreover, enzymatic methods are safer and milder than the chemical ones [3]. To date, numerous food protein hydrolysates endowed with a biological function, such as angiotensin I-converting enzyme (ACE) inhibition and mineral binding, as well as antidiabetic, satiating, immunomodulating, antioxidant or antimicrobial activities, have been reported [4] and several cryptides responsible for these activities have been identified. Blood pressure reduction activity is one of the most frequently observed health benefits in food protein cryptides [3,5].

Hypertension, which is a major cause of premature death worldwide, is a condition in which the blood vessels have persistently elevated pressure [6]. Angiotensin I-converting enzyme (ACE, EC 3.4.15.1) is a zinc-dependent carboxypeptidase that strongly influences the regulation of blood pressure by playing a double physiological role: it catalyzes the conversion of the decapeptide angiotensin I into the octapeptide angiotensin II, a potent vasopressor, and it promotes the inactivation of the vasodilator bradykinin [7]. ACE inhibitors, e.g., lisinopril, benazepril, enalapril and captopril, are the gold standard for treating hypertension [8]. However, the prolonged use of ACE inhibitor drugs is associated with negative side effects, such as dry cough, skin rashes, taste alterations, headache, fatigue, nausea and even hyperkalemia [4]. Therefore, food-derived protein hydrolysates with ACE-inhibitory activity have attracted a growing interest as potential ingredients in functional foods [9] to complement antihypertensive drugs, with the advantage of having fewer adverse effects [10]. ACE-inhibitory activity of protein hydrolysates can be influenced by several factors, such as protein source, hydrolysis conditions, degree of hydrolysis, molecular weights of generated peptides and their amino acid sequences [11].

Soybeans are one of the world’s most abundant sources of plant protein, with a rich and balanced amino acid profile. Moreover, it is well-recognized that soybean proteins are an important source of bioactive peptides (cryptides) [5] and that soy protein hydrolysates (SPH) represent an alternative to native proteins due to their superior functional properties [12]; in particular, increased solubility and nutritional characteristics. Specifically, SPH are reported to be suitable as food substitutes for the preparation of hospital diets, geriatric products, high-energy supplements and hypoallergenic infant formulas [3,12]. Soybean bioactive peptides are frequently endowed with antihypertensive activity [5], and although several ACE-inhibitory peptides with different chain lengths and amino acid compositions (e.g., NWGPLV, DLP, LSW, DG, GY, SY, etc.) have been identified in soybean protein hydrolysates, their mechanisms of action remain unclear [13,14]. Thus, further investigations are necessary, both to validate the antihypertensive potential of soybean hydrolysates and cryptides [15], and to overcome the challenges related to their production and purification processes in good yield [14,16]. 

The aim of this work is the development of an enzymatic process to achieve bioactive soybean hydrolysates as promising natural ingredients for nutraceutical applications. The use of the commercially available food-grade proteases Alcalase^®^ and Flavourzyme^®^ and their combination led to the preparation of soybean hydrolysates with a high degree of hydrolysis, good ACE-inhibitory activity and techno-functional properties suitable for scale-up. A novel ACE-inhibitory cryptide was identified, and its mechanism of action and stability in simulated gastrointestinal digestion were evaluated.

## 2. Materials and Methods

### 2.1. Materials

All solvents and reagents were purchased from Merck Life Science (Milano, Italy) and Fluorochem (Milano, Italy) and were used without further purification. Soy protein isolate (SPI) type 90 was kindly provided by ABS FOOD s.r.l. (Peraga di Vigonza, Italy). The chemical composition was ≥90.0% proteins, ≤7.0% moisture, ≤1.0% fat, ≤6.0% ash, and 1% total fiber (dry matter basis as reported by the producer). Proteolytic enzymes were kindly supplied by Novozymes^®^ (Bagsværd, Denmark): the endopeptidase Alcalase^®^ 2.4 L FG (enzyme activity: 2.4 AU g^−1^) derived from *Bacillus licheniformis* and the mixture of endopeptidase and exopeptidase Flavourzyme^®^ (enzyme activity: 1000 LAPU g^−1^) derived from *Aspergillus oryzae*.

### 2.2. Hydrolysis of Soy Protein Isolate (SPI)

Soy protein isolate (SPI) was suspended in distilled H_2_O (10% w_SPI_/v_H2O_) and the resulting mixture was heated to 50 °C, then treated with proteolytic enzyme(s) under mechanical stirring. Reaction conditions were set up according to the optimal pH and temperature for each enzyme [17,18]. In the case of one-enzyme hydrolysis, Alcalase^®^ was added (1% w/w_SPI_) after adjusting pH to ca. 8.5 with 0.1 M NaOH, whereas Flavourzyme^®^ was added (2% w/w_SPI_) after adjusting pH to ca. 6.5 with 0.1 M HCl, and the resulting mixtures were incubated for 24 h. In the two-enzyme experiment, proteins were initially hydrolyzed by Alcalase^®^ (1% w/w_SPI_) at pH ca. 8.5 for 2 h. The pH was then adjusted to ca. 6.5 and the mixture was further hydrolyzed by Flavourzyme^®^ (2% w/w_SPI_) for 22 h. The enzymes were inactivated by heat treatment at 85 °C for 15 min. Reaction mixtures were then centrifuged at 9000 rpm for 15 min, and supernatants (soy proteins hydrolysates, SPH) were collected, freeze-dried and stored at −20 °C. SPH yields for each enzymatic treatment (A, F, A + F) are reported in Table 1.

### 2.3. Protein and Hydrolysate Profiles

#### 2.3.1. Degree of Hydrolysis (DH)

The degree of hydrolysis (DH) of the obtained SPH (A, F, A + F) was determined using the trinitrobenzenesulfonic acid (TNBS) method [19,20]. Each assay was performed in triplicate. L-Leucine (0–2 mM) was used to generate a standard curve. DH values were calculated using the following equation [21]:$$\mathrm{DH}\;(\%)=100\;\left(\frac{{\mathrm{AN}}_{2}{-\mathrm{AN}}_{1}}{N_{\mathrm{pb}}}\right)$$
where AN_1_ is the amino nitrogen content of the protein substrate before hydrolysis (SPI, mg g^−1^_protein_), AN_2_ the amino nitrogen content of the protein substrate after hydrolysis (SPH, mg g^−1^_protein_), and N_pb_ the nitrogen content of the peptide bonds in the protein substrate (mg g^−1^_protein_). The nitrogen contents of both SPI and SPH were determined by elemental analysis using the value of 6.25 as protein conversion factor. A N_pb_ value of 88.01 mg g^−1^_protein_ was used for SPI. This value was obtained by assuming that all the nitrogen content in the starting material was related to proteins, and by multiplying the molecular mass of nitrogen by the moles sum of each amino acid present in the SPI, as reported in the amino acid profile provided by the supplier. Resulting DH% values are listed in Table 1.

#### 2.3.2. SDS-PAGE Analysis

The molecular weight distribution of SPI and of the three SPH (A, F, A + F) was determined by SDS PAGE analysis (Figure 1), using the NuPAGE^®^ Electrophoresis System (Invitrogen, Thermo Fisher, Milano, Italy) with a 12% pre-cast NuPAGE^®^ Novex^®^ Bis-Tris Gel. Prior to analysis, 65 μL of each sample (SPI concentration: 1 mg mL^−1^; SPH concentration: 10 mg mL^−^^1^) was mixed with 25 μL of NuPAGE^®^ LDS Sample Buffer (4×) and 10 μL of NuPAGE^®^ Reducing Agent (10×). Samples were heated at 70 °C for 10 min for denaturing proteins. Then, 10 μL of each sample were loaded in each well, and Thermo Scientific ^TM^ PageRuler ^TM^ Prestained Protein Ladder (M) was added as a molecular weight marker. Electrophoresis was performed using the NuPAGE^®^ MES SDS Running Buffer (pH 7.3), in an *X*Cell *Surelock ^TM^* Mini-Cell, at constant voltage (200 V; 1 h). The gel was stained with 0.1% Comassie R-250 in 40% ethanol and 10% acetic acid solution and destained in 10% ethanol and 7.5% acetic acid solution.

### 2.4. Ultrafiltration

Each SPH was dissolved in milli-Q H_2_O, and the resulting solutions (*ca* 1% *w*/*v*) were ultrafiltered through 10, 5 and 1 kDa molecular weight cut-off membranes (MWCO) by using an Amicon^®^ ultrafiltration cell (model 8400, 400 mL, Merck Millipore Corporation, Milano, Italy). Fractions ranging from 10 kDa to 1 kDa were recovered separately and freeze-dried. Ultrafiltration yields (*w*/*w*,%, referring to the weight of the starting dry hydrolysates) are reported in Figure 2.

### 2.5. Characterization of SPH < 1 kDa Fraction

#### 2.5.1. ACE Inhibition Assay

ACE-inhibitory activity was determined as previously reported [22], by evaluating hippuric acid (HA) formation from hippuryl-histidyl-leucine (HHL), a mimic substrate for angiotensin I, by HPLC analysis in the absence of and at increasing concentrations of the inhibitor (86.3, 172.5, 345.1, 690.1, 1035.2 μg mL^−1^). IC_50_ is defined as the concentration of the inhibitor required to inhibit 50% of the ACE activity under the assay conditions. IC_50_ values are reported in Table 2 as mean value ± standard deviation (SD) of three independent assays.

#### 2.5.2. Determination of ACE Inhibition Pattern 

The inhibition mode by the peptide showing the highest ACE-inhibitory activity (i.e., NDRP) was determined using Lineweaver–Burk plots (Figure 3) of 1/HA production rate vs. 1/HHL concentration (mM^−1^). ACE inhibition was measured by varying the concentration of the enzyme substrate HHL (2.32, 1.16, 0.58, 0.29 mM) in the absence of the inhibitor and using two different concentrations of the peptide (0.689, 0.172 mM). All experiments were performed in triplicate.

#### 2.5.3. RP-HPLC

RP-HPLC experiments were performed on an AKTA Basic 100 instrument (Cytiva Europe GmbH, Buccinasco, Italy), equipped with a UV detector and monitored by the Unicorn ™ Software (Cytiva Europe GmbH, Buccinasco, Italy). Analytical and semipreparative analyses were carried out on reverse phase Jupiter 10 μm Proteo 90 Å columns (250 mm × 4.6 mm and 250 mm × 10 mm, respectively) (Phenomenex, Torrence, CA, USA), at room temperature and at a flow rate of 1 mL min^−1^ and 5 mL min^−1^, respectively. Samples were dissolved in 0.1% formic acid in water. 

Chromatographic conditions: 0.1% formic acid in water (solvent A) and 0.1% formic acid in water/acetonitrile, 2:8 (solvent B); from 0% to 30% B in 30 min, then to 100% B in 10 min; detection at λ 226 nm. Semipreparative separation (see Appendix A) of SPH with MW < 1 kDa obtained by hydrolysis with Alcalase^®^ (A < 1 kDa) afforded three fractions (A1, A2 and A3) that, after removal of the organic solvent under reduced pressure, were freeze dried and stored at −20 °C.

#### 2.5.4. NMR Analysis

NMR experiments were performed at 298 K on 400 MHz Bruker NMR spectrometer (Bruker Corporation, Billerica, MA, USA) equipped with a z-gradient coil probe. Chemical shifts (δ) are given in parts per million (ppm) and were referenced to the solvent signal (DMSO-*d_6_*, δ_H_ 2.50 ppm and δ_C_ 39.52 ppm from TMS (tetramethylsilane), respectively). All 1D and 2D NMR spectra were collected using the standard pulse sequences available with Bruker Topspin 3.6. The 2D TOCSY and 2D NOESY analyses were recorded with short mixing times: 80 ms and 500 ms, respectively. Proton resonances were assigned using standard methods. See Appendix A.

#### 2.5.5. Determination of Amino Acid Sequence 

For amino acid sequence determination, fraction A1 was submitted to automatic Edman degradation (Hewlett-Packard Protein Sequencer HP G1000A, Agilent Technologies Italia S.p.A., Cernusco sul Naviglio, Milano, Italy), using the manufacturer’s phenylisothiocyanate (PTH)-4M HPLC method (Hewlett-Packard Technical note 95-1). In detail, the sample was loaded on a Hewlett-Packard adsorptive biphasic column, and the PTH-derivatized amino acids were identified by comparison with a reference mixture of phenylthiohydantoin amino acids. Analysis of the relative abundance of the PTH-amino acid peaks in the chromatograms obtained after each cycle of Edman degradation (see Appendix A) led to the identification of putative sequences, which were used as queries for the screening of the soybean protein sequences in the Uniprot database [23].

The main components of the A1 fraction were found to be a heptapeptide, whose putative sequence was ALKPDNR, and three tetrapeptides, whose putative sequences were VVPD, NDRP and NDTP, respectively. All the peptide patterns were searched against the UniProt database [23] and were found to be present in soybean glycinins (G1, G2, G3).

#### 2.5.6. Mass Spectrometry of Synthetic Peptides

Electrospray ionization mass spectra (ESI-MS) were recorded on a Thermo Finnigan LCQ Advantage spectrometer (Hemel Hempstead, Hertfordshire, UK). Mass spectra acquisition was performed in positive and negative ion mode in a mass range of 50–2000 *m*/*z*. Source parameters: T, 200 °C; sheath gas flow rate, 20 arb; aux gas flow rate, 10 arb; spray voltage, 5.50 kV; capillary temperature, 275 °C; capillary voltage, 38.00 V; tube lens, 65.00 V.

Electrospray ionization high-resolution mass spectrometry (ESI-HR-MS) analyses were performed on a Synapt G2-Si QTof instrument (Waters, Milford, MA, USA) equipped with a ZsprayTM ESI-probe for electrospray ionization (Waters, Milford, MA, USA). The sample was dissolved in water–0.1% formic acid in methanol and analyzed by direct infusion. The ion source and interface conditions were as follows: capillary voltage 1.3 kV, sampling cone 120 V, source temperature 120 °C, desolvation temperature 150 °C and desolvation gas flow rate 600 L h^−1^. Mass spectra were acquired over the *m*/*z* range 50–1000 in a positive ion Full Scan mode. Leucine enkephalin was used as a lock-mass compound. Data were processed with MassLynxTM V4.2 software (Waters Milford, MA, USA).

### 2.6. Peptides Synthesis 

The peptides identified by the Edman degradation (ALKPDNR, VVPD, NDRP and NDTP) were synthesized by microwave-assisted solid phase automated synthesis using the Fmoc-protocol on a Biotage^®^ Initiator^+^ SP wave peptide synthesizer (Biotage, Uppsala, Sweden). All the syntheses were carried out on 200 mg of preloaded 2-chlorotrityl resin, corresponding to 0.18 mmol of Fmoc-Arg(Pbf)- used for ALKPDNR, 0.14 mmol of Fmoc-Asp(OtBu)- for VVPD and 0.13 mmol of Fmoc-Pro- for NDRP and NDTP, respectively. Fmoc deprotection was performed using 25% piperidine in DMF. Each coupling reaction was carried out at 50 °C in DMF using HOBt and HBTU (3-fold excess) as coupling agents and DIPEA (6-fold excess) as a base; reaction time was 15 min. The functional groups of the amino acid side chains were protected as follows: Asn(Trt), Asp(O*t*Bu), Lys(Boc), Arg(Pbf) and Thr(*t*Bu). Each amino acid was used in 3-fold excess and dissolved along with coupling agents and DIPEA in DMF (3 mL) 15 min before reaction. For cleavage, a trifluoroacetic acid (TFA) (15.60 mL)/phenol (0.89 g)/H_2_O (0.88 mL)/triisopropylsilane (TIPS) (0.34 mL) solution was used; all resins were treated with the cleavage solution for 1–5 h, depending on the presence of the Trt protecting group at the *N*- terminus of the peptide, which takes longer to be cleaved. The solid supports were then removed by filtration and the resulting yellow oils were treated with Et_2_O at 0 °C. The white precipitates were centrifuged (4000 rpm, 10 min), decanted, dissolved in H_2_O and freeze-dried. 

The identity and molecular weight of the obtained peptides (ALKPDNR, 19%; VVPD, 51%; NDRP, 45%; and NDTP, 31% yield, respectively) were confirmed by ESI-MS in positive ion mode analysis.

**ALKPDNR ESI-MS:***m*/*z* calcd. for [C_34_H_61_N_12_O_11_]^+^: 813.45, found: 813.32 [M + H]^+^, 835.35 [M + Na]^+^.

**VVPD ESI-MS:***m*/*z* calcd. for [C_19_H_33_N_4_O_7_]^+^: 429.23, found: 429.53 [M + H]^+^, 451.67 [M + Na]^+^, 857.05 [2M + H]^+^, 879.23 [2M + Na]^+^.

**NDRP ESI-MS:***m*/*z* calcd. for [C_19_H_33_N_8_O_8_]^+^: 501.23, found: 501.56 [M + H]^+^, 523.50 [M + Na]^+^, 1001.47 [2M + H]^+^.

**NDTP ESI-MS:***m*/*z* calcd. for [C_17_H_28_N_5_O_9_]^+^: 446.18, found: 446.32 [M + H]^+^, 468.46 [M + Na]^+^, 891.21 [2M + H]^+^, 913.18 [2M + Na]^+^.

Moreover, the peptide identified as the most active ACE inhibitor, namely NDRP, was further characterized by ^1^H and ^13^C NMR analysis and HR-MS.

**^1^****H NMR (400 MHz, DMSO-*d*_6_)****δ****/*ppm*:** 8.71 (d, J = 7.4 Hz, 1H), 8.10 (d, J = 7.7 Hz, 1H), 7.79 (s, 1H), 7.70 (s, 1H), 7.23 (br s, 2H), 4.57 (m, 1H), 4.45 (m, 1H), 4.24 (dd, J = 8.6, 4.5 Hz, 1H), 4.05 (dd, J = 8.2, 4.5 Hz, 1H), 3.64 (m, 1H), 3.53 (m’, 2H), 3.10 (br d, J = 5.1 Hz, 2H), 2.70 (ddd, J = 25.3, 16.8, 4.4 Hz, 2H), 2.57 (dd, J = 14.4, 5.7 Hz, 2H), 2.15 (m, 1H), 1.96–1.77 (m, 3H), 1.71 (br s, 1H), 1.56 (br s, 2H).

**^13^****C NMR (100 MHz, DMSO-*d*_6_)****δ****/*ppm*:** 173.67, 171.99, 171.15, 170.55, 169.89, 168.94, 158.90, 158.59, 157.33, 59.03, 50.67, 50.03, 49.59, 46.86, 40.97, 36.61, 36.22, 29.07, 28.37, 25.01, 24.91.

**NDRP HR-MS****:***m*/*z* calcd. for [C_19_H_33_N_8_O_8_]^+^: calcd: 501.2421, found: 501.2413.

### 2.7. Detection of Peptide NDRP in the A < 1 kDa Fraction by Multiple Reaction Monitoring (MRM)

Pure NDRP obtained by chemical synthesis and the hydrolysate A < 1 kDa were analyzed on an SL IT mass spectrometer interfaced with a HPLC-Chip Cube source (Agilent Technologies, Palo Alto, CA, USA). Data were processed with MSD Trap control 4.2, and Data analysis 4.2 version (Agilent Technologies, Palo Alto, CA, USA). The samples (1 µL), acidified with formic acid, were injected in infusion mode. Data acquisition occurred in positive ionization mode. Capillary voltage was −1900 V, with endplate offset −500 V. Mass spectra were acquired with an ICC target of 30,000 and a maximum accumulation time of 150 ms. LC/MS analysis was performed in multiple reaction monitoring (MRM) mode. Specifically, the detection of NDRP was carried out by monitoring the mono-charged precursor ion [M + H]^+^ (*m*/*z* 501.2). The MS/MS fragment ions were then covered along the peptide sequence. 

### 2.8. In Vitro Simulated Digestion of NDRP

The in vitro simulated digestion of NDRP was carried out according to the method reported by Xu [14], with slight modifications. Briefly, NDRP (49.9 mg) was dissolved in HPLC-grade H_2_O (10 mg mL^−1^) and the pH of the solution was adjusted to 2 with 0.1 M HCl. Pepsin (1 mg) was added, and the mixture was stirred at 37 °C for 2 h. Then, the pH was adjusted to 6 with 0.9 M NaHCO_3_, and afterwards to 7.5 with 1 M NaOH. Pancreatin (1 mg) was added, and the solution was incubated at 37 °C for 4 h. After deactivation of the enzymes by heating in a boiling water bath for 10 min, the solution was cooled and freeze-dried, and the resulting in vitro-digested NDRP (55.6 mg, 10% of salts) was submitted to ESI-MS and ACE-inhibitory activity assay (ACE inhibition = 82.47 ± 1.07%).

**NDRP after in vitro simulated digestion ESI-MS**: (*m*/*z* calcd. for [C_19_H_33_N_8_O_8_]^+^: 501.23 Da, found: 501.55 Da [M + H]^+^, 523.51 Da [M + Na]^+^).

### 2.9. Data Treatment and Statistical Analysis 

Degrees of hydrolysis (DH) was calculated by taking into account the propagation of the experimental error; the significance of the results is expressed through the evaluation of discrepancies. Statistical analyses of ACE-inhibitory activities were performed with StatGraphics Plus (version 2.1 for Windows). The data were evaluated using one-way analysis of variance followed by a Fisher’s Least Significant Difference procedure; values with different letters are significantly different for *p* < 0.05. 

## 3. Results and Discussion

### 3.1. Enzymatic Hydrolysis of Soy Proteins and Degree of Hydrolysis

Among the numerous proteases commonly employed to hydrolyze soybean protein [24], we selected Alcalase^®^ and Flavourzyme^®^. Alcalase^®^, a mixture of endopeptidases, has been found to be one of the most efficient biocatalysts in releasing bioactive peptides from different protein sources [25]. Flavourzyme^®^, a mixture of endopeptidases and exopeptidases, is frequently used for debittering food protein hydrolysates [26]. In this work, these proteolytic enzymes were used individually and in combination. 

A commercial soy protein isolate (SPI) was hydrolyzed by treatment with Alcalase^®^ and Flavourzyme^®^ separately and by sequential treatment with both Alcalase^®^ and Flavourzyme^®^. Table 1 reports the rough weight percentages of the three different SPH (A, F and A + F) obtained after centrifugation and lyophilization of supernatants, with respect to the amount of starting SPI. The enzyme combination appeared to be more effective in promoting protein hydrolysis, considering that the amounts of hydrolysates (as dry weight% *w*/*w*) recovered, i.e., A, F and A + F, were 43%, 64% and 77%, respectively (Table 1).

Moreover, Table 1 reports the degree of hydrolysis (DH) values of the three hydrolysates (A, F, A + F). DH is defined as the proportion of cleaved peptide bonds in a protein hydrolysate, and is routinely used to compare different proteolytic processes. The enzyme choice, according to its specificity and selectivity, represents a crucial parameter to tune the DH and the final properties of the hydrolysates [25,27]. The DH of the obtained SPH (A, F, A + F) was determined using the trinitrobenzenesulfonic acid (TNBS) method [19], which is independent from the type of enzyme activity used during hydrolysis, unlike other reported methods [21]. The sample hydrolyzed by Flavourzyme^®^ and the sample hydrolyzed by a combination of Alcalase^®^ and Flavourzyme^®^ showed similar and not statistically different DH (34 ± 5.5% and 36 ± 5.6%, respectively), while the hydrolysate prepared with Alcalase^®^ alone had the lowest degree of hydrolysis (17 ± 5.3%). Thus, the combination of A + F exhibited nearly the same DH as for F used individually, while allowing an increase in hydrolysate recovery by more than 10% (Table 1). This result could be attributable to the operative conditions (pH and reaction time) selected for the experiment carried out with the enzymes in combination [11]. In fact, when combining enzymes, optimum conditions for each of them may not be attained. In addition, the latter enzyme hydrolyzes the peptides cleaved by the former biocatalyst.

### 3.2. SDS-PAGE

All three hydrolysates, as well as untreated SPI, were submitted to SDS-PAGE in NuPAGE^®^ MES SDS Running Buffer (pH 7.3), using a 12% pre-cast NuPAGE^®^ Novex^®^ Bis-Tris Gel, which was selected for resolving proteins/peptides in the range 1–200 kDa. The resulting profiles are shown in Figure 1. Lanes 2–5 represent SPI, A, F and A + F, respectively. SPI (lane 2) presents an electrophoretic pattern characteristic for soy proteins [26]: the highly intense bands at 72 kDa and about 67 kDa and the less intense one at 63 kDa are attributable to α’, α and β subunits of β-conglycinin, respectively. Furthermore, the acidic chain A3 of the G1 subunit of glycinin is observed at about 33 kDa and the basic chain of subunit G2 at 22 kDa [28]. It is worth noting that the use of both Alcalase^®^ and Flavourzyme^®^, as well as of their combination, led to the complete disappearance of the bands of the main proteins commonly found in soy. In particular, after hydrolysis with Alcalase^®^ (lane 3), the greater part of the hydrolysate had a molecular weight between 12 and 20 kDa; by contrast, the only band slightly visible after reaction with Flavourzyme^®^ (lane 4) is the one at about 14 kDa. In addition, hydrolysis by Alcalase^®^ + Flavourzyme^®^ resulted in undetectable peptides with molecular weight higher than 10 kDa, thus indicating that such a protease mixture produced very small peptides. This pattern was consistent with results obtained from DH and ultrafiltration (see Section 3.1 and Section 3.3, respectively).

### 3.3. Ultrafiltration and Evaluation of ACE-Inhibitory Activity

Results of SDS-PAGE and DH analysis suggested that SPHs produced following the three hydrolytic protocols are extensively hydrolyzed and composed of very small peptides, having a molecular weight lower than 15 kDa. The three SPHs (A, F, A + F) were fractionated by ultrafiltration through decreasing molecular weight cut-off membranes (10, 5 and 1 kDa). 

As shown in Figure 2, F and A + F mainly contained < 1 kDa peptides (72% and 76%, respectively). Conversely, in the reaction with Alcalase^®^, the peptide fraction < 1 kDa represented only 42%, whereas higher amounts of peptides with a MW in the range 1–5 kDa and with MW > 10 kDa were produced (25% and 27%, respectively). The peptide size distribution of the protein hydrolysates depended on the protease mixture selected to perform proteolysis, and our results are in good agreement with the well-known mechanism of action of the two enzyme preparations (Alcalase^®^ and Flavourzyme^®^) [27]. Indeed, the action of exopeptidases is facilitated by a previous hydrolytic step carried out by endopeptidases: firstly, endopeptidases of Alcalase^®^ and Flavourzyme^®^ hydrolyze peptide bonds within soy proteins, producing relatively large peptides, then exopeptidases of Flavourzyme^®^ (two aminopeptidases and two dipeptidyl peptidases) [29] systematically remove amino acids or very small peptides from either the *N*-terminus or the *C*-terminus position, thus resulting in an increased hydrolytic degradation.

Short chain peptides are known to be generally good candidates as ACE inhibitors [30]. Thus, the three fractions with MW less than 1 kDa that were prepared by sequential ultrafiltration of A, F and A + F (A < 1 kDa, F < 1 kDa, A + F < 1 kDa) were investigated for their ACE-inhibitory activity. The results obtained are shown in Table 2. Fraction A < 1 kDa, obtained by Alcalase^®^ treatment, exhibited the maximal ACE-inhibitory activity (77.01 ± 0.57%) at the highest tested concentration of hydrolysate. The fraction arising from hydrolysis with Alcalase^®^ and Flavourzyme^®^ reached 66.54 ± 0.59% inhibitory activity at the same concentration, whereas the activity of F < 1 kDa fraction dropped to 54.42 ± 3.34%.

### 3.4. Purification, Characterization and Synthesis of ACE-Inhibitory Peptides

The most active fraction, i.e., A < 1 kDa, was selected and separated by preparative RP-HPLC into three subfractions (A1, A2 and A3), which were in turn screened for their ACE-inhibitory activities (see Table 2 and Appendix A). The data in Table 2 indicate that the purification step obtained a peptide mixture with improved antihypertensive properties, considering that in the isolated fraction A1, the ACE inhibition activity increased from 77.01 ± 0.57% to 92.2 ± 0.13% with an IC_50_ value of 231.75 ± 2.43 μg mL^−1^. 

With the aim of determining the composition of the A1 fraction, NMR and MS analyses were carried out. ^1^H NMR spectrum in DMSO-*d_6_* appeared to be very crowded, especially in the amide-NH region between 6.5 and 9.0 ppm, indicating that the sample was a mixture of many different peptides. In order to obtain more structural information, 2D spectra were collected. Complete sequence-specific assignment of resonances of some amino acids was achieved using a combination of COSY and TOCSY experiments [31]. In particular, the spin systems of alanine (A), valine (V), aspartic acid (D) and arginine (R) (see Appendix A) were evidenced from the analysis of the NH-αH region and that of proline (P) from the analysis of the αH-αH region of the TOCSY spectrum. Moreover, no near-neighbour connections were found in the NOESY and ROESY spectra, thus suggesting that these extremely short peptides are not able to adopt a folded structure for a significant fraction of the time [32].

As spectroscopic analysis did not allow the identification of peptides in the isolated A1 fraction, it was submitted to Edman degradation, an effective method for determining the sequence of unknown peptides with a free amino terminus. Four putative sequences were identified; in particular, one heptapeptide (ALKPDNR) and three tetrapeptides (VVPD, NDRP and NDTP). It should be pointed out that all these peptide patterns are present within the sequences of storage soybean proteins (glycinins G1, G2 and G3), according to the UniProt database [23].

In order to define which peptide/s is/are mainly responsible for the ACE-inhibitory properties exhibited by A1 fraction, all identified peptides were synthesized by solid phase automated peptide synthesis using the Fmoc protocol, and their ACE-inhibitory activities were evaluated (Table 2). Among them, only NDRP showed an ACE inhibition capacity comparable to that of the starting A1 fraction (maximal ACE-inhibitory activity = 87.76%, IC_50_ = 148.28 µg mL^−1^) (Table 2). This finding suggested that this peptide might be the main agent responsible for the observed bioactivity.

### 3.5. Determination of the ACE Inhibition Mechanism

To investigate the inhibition mechanism, Lineweaver–Burk plots of ACE activity in the presence and absence of NDRP were drawn (Figure 3). These plots, characterized by a coincident intercept on the 1/S axis, revealed that the peptide acts as a non-competitive inhibitor [33].

Inhibition is described as non-competitive when the inhibitor molecule has the same affinity for both the enzyme and enzyme–substrate complex. Hence, the peptide NDRP does not bind to ACE catalytic site, but to other sites, resulting in a decreased efficacy of the enzyme, independently of substrate binding [20].

Although competitive ACE inhibitors have been the most frequently reported, numerous peptides showing remarkable non-competitive ACE-inhibitory activity have also been identified, from both plant and animal food proteins. Some of them are: IFL (IC_50_ of 44.8 μM) and WL (IC_50_ of 29.9 μM) from Tofuyo (fermented soybean food) [34]; DENSKF (IC_50_ of 100 μM) from fruits of *T. chebula* Retz. [33]; RYPSYG (IC_50_ of 54 μg mL^−1^) and DERF (IC_50_ of 21 μg mL^−1^) from bovine casein hydrolysate [35]; GDLGKTTTVSNWSPPKYKDTP (IC_50_ of 11.28 μM) from tuna frame protein hydrolysate [7]; RLPSEFDLSAFLRA (IC_50_ of 0.45 mg mL^−1^) and RLSGQTIEVTSEYLFRH (IC_50_ of 1.10 mg mL^−1^) from *Pleurotus cornucopiae* mushroom [36]. To the best of our knowledge, no non-competitive ACE-inhibitory peptide has been identified in soybean protein isolates so far. As non-competitive ACE-inhibitory peptides bind to a different site from the substrate, their structure–activity relationships are not yet well established.

Usually, ACE-inhibitory peptides have a short chain bearing 2–12 residues, although active peptides with up to 21 amino acids have been identified [7,37]. Concerning amino acid composition and peptide sequence, the presence of a net negative charge induced by highly acidic amino acids (Asp and Glu) may reduce the catalytic rate of ACE by chelating zinc atom, which is vital for ACE activity [10]. Moreover, it has been reported that the presence of tyrosine, phenylalanine, tryptophan, proline, lysine, isoleucine, valine, leucine and arginine in peptides has a strong influence on ACE binding. In particular, most ACE-inhibitory peptides carry hydrophobic or branched amino acids at the *N*-terminus, while residues with cyclic or aromatic rings, such as tyrosine, phenylalanine, tryptophan and proline, are characteristic for the *C*-terminus [22,37]. The non-competitive ACE inhibitor peptide isolated during this study (NDRP) possesses a few features that are consistent with its ACE-inhibitory activity: it is short (it is composed of four amino acids), it bears an acidic amino acid (Asp) as a potential zinc-chelating site, and a Pro residue at the *C*-terminus.

### 3.6. Detection of Peptide NDRP in the A < 1 kDa Fraction Assessed by MRM Mass Spectrometry Analysis

To confirm the presence of NDRP in the A < 1 kDa fraction, the multiple reaction monitoring (MRM) technique was employed using an ion trap mass spectrometer by infusion. Specifically, NDRP was observed by monitoring the mono-charged precursor ion [M + H]^+^ (*m*/*z* 501.2). The MS/MS fragment ions were then covered along the peptide sequence. Figure 4 shows the MS/MS spectrum of NDRP with the highlighted sequence coverage. The ions *y*_3_ and *y*_2_, corresponding to DRP and RP, respectively, were the main detected ions, whereas the *b*_2_ ion, corresponding to ND, was the most representative ion of the *b* series.

### 3.7. Stability of NDRP toward In Vitro Simulated Digestion

To exert their health-promoting properties, bioactive peptides must survive gastrointestinal digestion and reach their target sites in an intact and active form. It is known that short-chain peptides, especially those with *C*-terminal proline, are generally less susceptible to degradation by proteolytic enzymes [13,38]. Moreover, they are absorbed in the small intestine more rapidly than free amino acids and large oligopeptides [30].

To evaluate the resistance of NDRP to digestive enzymes, this peptide was subjected to a two-stage hydrolysis process that simulated in vivo conditions during physiological digestion, and the changes in its bioactivity were examined. After treatment with gastrointestinal proteases (pepsin and pancreatin), the maximal ACE inhibition of the NDRP digested sample was found to be 82.47 ± 1.07%. This value roughly corresponds to about 94% of the ACE-inhibitory activity of pure NDRP, thus indicating that NDRP might be resistant to gastrointestinal digestion. In addition, the in vitro-digested NDRP was analyzed by ESI-MS in positive ion mode, and the presence in the spectrum of a peak corresponding to the molecular weight of the peptide (*m*/*z* calcd. for [C_19_H_33_N_8_O_8_]^+^: 501.23 Da, found: 501.55 Da [M + H]^+^, 523.51 Da [M + Na]^+^) was observed, further corroborating its stability in intestinal proteolysis. 

It is worth noting that in a recent study conducted by Xu and co-workers [14], a number of different peptides were detected in a hydrolysate of soy protein prepared by Alcalase^®^ in experimental conditions different from those used in this work (pH = 7.5, 55 °C, 8 h vs. pH = 8.5, 50 °C, 24 h). Differences in the hydrolytic conditions as well as in the starting material and separation techniques may affect the release and/or the identification of specific peptide sequences having different mechanisms of action. Indeed, all the peptides reported by Xu are competitive ACE inhibitors, and some of them were found to be unstable in in vitro simulated gastrointestinal digestion.

## 4. Conclusions

This study provides a comparative analysis of three hydrolytic protocols applied on inexpensive soy protein isolate, exploiting cheap and commercially available proteolytic enzymes (Alcalase^®^ and Flavourzyme^®^) used either individually or in combination. The resulting soy protein hydrolysates (A, F and A + F) consisted of small peptides with enhanced solubility in water, with respect to starting proteins.

Furthermore, the hydrolysate obtained through the process catalyzed by Alcalase^®^ (A) was found to have the highest ACE-inhibitory activity. From this hydrolysate, upon a sequential ultrafiltration process, a novel peptide (NDRP) was identified and prepared by chemical synthesis. To the best of our knowledge, such a peptide has never been isolated from either soy and soybean-derived products or from other food proteins. Besides ACE-inhibitory activity, NDRP was found to be highly stable in in vitro simulated gastrointestinal digestion (residual activity of the digested sample was 94% of the ACE-inhibitory activity of the pure peptide).

Our data confirm the relevance of soy as a valuable and cheap source of bioactive peptides that may be exploited as components of functional foods for the prevention of chronic diseases.

## Figures and Tables

**Figure 1 foods-11-02667-f001:**
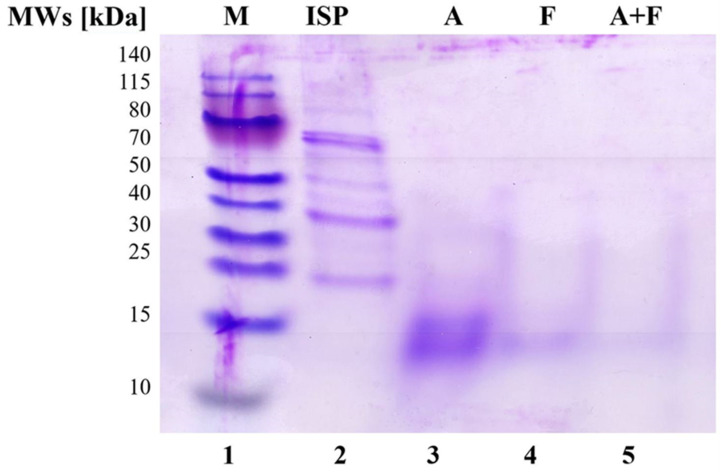
SDS-PAGE analysis of protein MWs by 12% precast Bis-Tris Invitrogen^®^: Thermo Scientific ^TM^ PageRuler ^TM^. Lanes: prestained protein ladder (M, lane 1); soy protein isolate (SPI, lane 2); SPH obtained from the reaction with Alcalase^®^ (A, lane 3), Flavourzyme^®^ (F, lane 4) and Alcalase^®^ + Flavourzyme^®^ (A + F, lane 5).

**Figure 2 foods-11-02667-f002:**
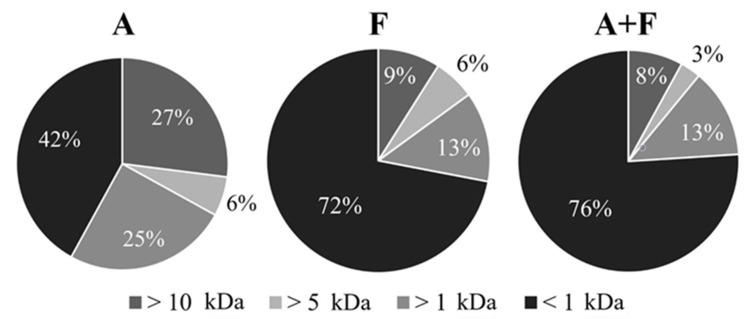
Yields (*w*/*w*,%) after sequential ultrafiltration of hydrolysates A, F, and A + F through MWCO membranes of 10, 5 and 1 kDa.

**Figure 3 foods-11-02667-f003:**
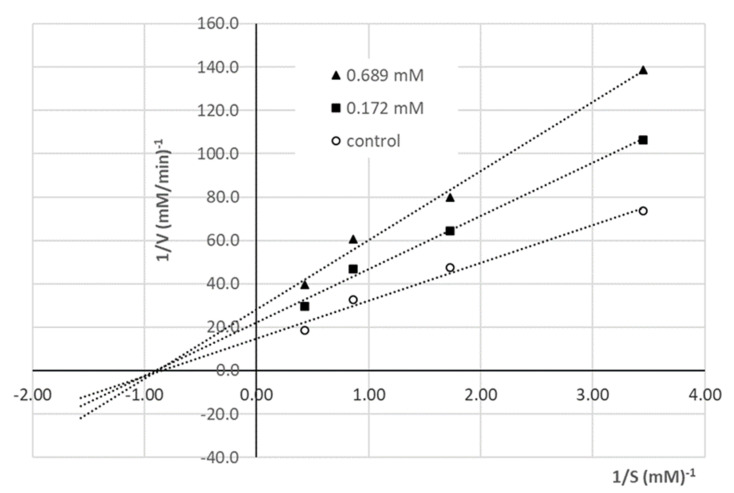
Lineweaver–Burk plots of ACE activity in the presence of NDRP at concentrations of 0.689 mM (▲) and 0.172 mM (■) and without the inhibitory peptide (○, control); HHL was used as the enzyme substrate. All experiments were conducted in triplicate.

**Figure 4 foods-11-02667-f004:**
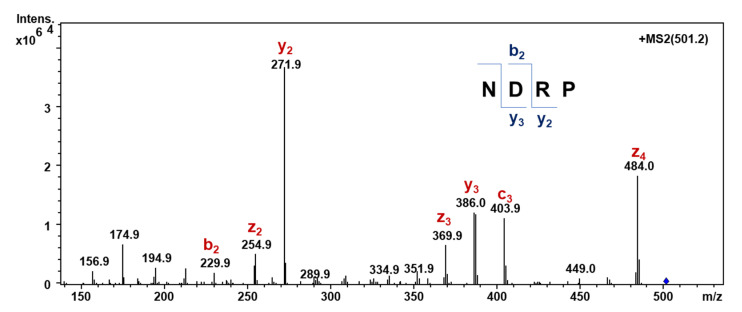
MS/MS spectrum of NDRP detected by infusion of A < 1 kDa by MRM.

**Table 1 foods-11-02667-t001:** Percentages of soy protein hydrolysates (SPH) after hydrolysis with Alcalase^®^, Flavourzyme^®^ and Alcalase^®^ + Flavourzyme^®^ (A, F and A + F, respectively) and degree of hydrolysis (DH).

Enzymatic Protocol	SPH (%)	DH (%) *
A	43	17 ± 5.3 ^a^
F	64	34 ± 5.5 ^b^
A + F	77	36 ± 5.6 ^b^

* Values are reported as mean ± error% of three independent replicates. Different superscript lowercase letters represent values with a discrepancy greater than the calculated error.

**Table 2 foods-11-02667-t002:** Angiotensin-Ⅰ-converting enzyme (ACE) inhibitory activity of three SPH < 1 kDa (A < 1 kDa; F < 1 kDa; A + F < 1 kDa), of three isolated RP-HPLC fractions from A < 1 kDa (A1; A2; A3) and of four peptides prepared by chemical synthesis.

Sample	Max ACE Inhibition (%) *	IC_50_ (μg mL^−1^) *
A < 1 kDa	77.01 ± 0.57 ^g^	296.57 ± 2.24 ^c^
F < 1 kDa	54.42 ± 3.34 ^d^	869.87 ± 15.46 ^f^
A + F < 1 kDa	66.54 ± 0.59 ^e^	558.35 ± 26.88 ^e^
A1	92.2 ± 0.13 ^i^	231.75 ± 2.43 ^b^
A2	70.63 ± 0.32 ^f^	361.66 ± 2.73 ^d^
A3	75.72 ± 0.18 ^g^	304.61 ± 4.81 ^c^
ALKPDNR	25.72 ± 0.84 ^b^	//
VVPD	13.23 ± 1.18 ^a^	//
NDRP	87.76 ± 0.61 ^h^	148.28 ± 9.83 ^a^
NDTP	46.15 ± 1.72 ^c^	//

* Values are reported as mean value ± standard deviation of three independent experiments; different superscript lowercase letters represent statistically significant differences between mean values at *p* < 0.05 obtained by one-way ANOVA followed by a Fisher’s Least Significant Difference procedure.

## Data Availability

Data is contained within the article or Appendix A.

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
