# Peer review of "Preparation, Characterization and In Vitro Stability of a Novel ACE-Inhibitory Peptide from Soybean Protein"

_foods, 2022, doi:10.3390/foods11172667_

Round 1

Reviewer 1 Report

In this study, the author provides a comparative analysis of three hydrolytic protocols applied on inexpensive soy protein isolate, exploiting cheap and commercially available proteolytic enzymes (Alcalase® and Flavourzyme®) used either individually or in combination. This study will contribute to the relevance of soy as a valuable and cheap source of bioactive peptides which may be exploited as components of functional foods for the prevention of chronic diseases. Overall, I feel that this paper is suitable for the area of Foods. The finding of this manuscript is interesting and I suggest revisions:

1)    Line 89: Why did you choose 50 °C, and is there any reference to it?

2)    Lines 92-93: The concentration of Alcalase® is 1 % (w/w SPI), whereas the con concentration of Flavourzyme® is 2 % (w/w SPI). How to compare the effects of different concentrations?

3)    Lines 269-281: The description in this part is unnecessary, it’s more like the introduction.

4)    Lines 282-284: This section is more like materials and methods than results and discussion.

5)    The data in this manuscript were not analyzed for significance i.e. table 1 & table 2, please add it.

6)    Lines 334-335: Emulsifying and 334 foaming properties? I have not seen this index in this manuscript.

7)    Lines 339-340: Why is the < 1 kDa peptides content of F higher than that of A+F, please explain it.

Author Response

Referee 1

We would like to thank the Reviewer for his/her insightful comments that help improve the manuscript.

1) Line 89: Why did you choose 50 °C, and is there any reference to it?

The working temperature for both Alcalase and Flavourzyme ranges approximately from 40 to 70°C. 50°C is frequently reported in the literature as optimum temperature in the hydrolysis of plant protein with both enzymes (for example references 17 and 18 of the revised manuscript).

2) Lines 92-93: The concentration of Alcalase® is 1 % (w/w SPI), whereas the con concentration of Flavourzyme® is 2 % (w/w SPI). How to compare the effects of different concentrations?

At the beginning of our work, as a first approach, a preliminary screening was performed with the aim of finding the optimized reaction conditions to obtain soy protein hydrolysates, mainly composed of biologically active small peptides, in high yields. When Alcalase was used as a biocatalyst for the enzymatic hydrolysis, the best results were obtained at 1% concentration (w/w SPI). By contrast, the use of the same enzyme/substrate ratio of Flavourzyme led to quite low yield. For this reason, we doubled its concentration. However, these proteolytic enzymes may have different activity towards the same substrate. As an example, see Food Bioprocess Technol 2011, 4, 1399–1406 where the comparison of the hydrolytic activity of Alcalase and Flavourzyme towards casein (the most used protein substrate for determining the specific activity under neutral and alkaline conditions) is reported.

3) Lines 269-281: The description in this part is unnecessary, it’s more like the introduction.

We agree with the referee and thus we have reduced the description in this part of the manuscript. The text has been partially moved in the introduction (lines 62-67 of the revised manuscript) as suggested.

4) Lines 282-284: This section is more like materials and methods than results and discussion.

The concentrations of proteases have been removed.

5) The data in this manuscript were not analyzed for significance i.e. table 1 & table 2, please add it.

Analysis of data for significance has been added. As the degree of hydrolysis (DH) for the three fractions (A, F, A+F) was calculated taking into account the propagation of the experimental error, significance was assessed by evaluating the discrepancy. ACE-inhibition data were subjected to ANOVA analysis followed by Fisher’s Least Significant Difference post-hoc test. Results have been included in Table 1 and in Table 2 and tables footnotes were corrected accordingly. Section 2.9 in the manuscript has been amended to include these modifications.

6) Lines 334-335: Emulsifying and 334 foaming properties? I have not seen this index in this manuscript.

The sentence referring to the emulsifying and foaming properties of the hydrolysates has been erroneously inserted in the text. It has been deleted in the revised manuscript. We apologise for this mistake due to the fact that our hydrolysates seem to possess functional properties (evaluated as emulsifying and foaming capacity) suitable for a possible scale-up. However, these issues have not been further deepened because outside the scope of our research.

7) Lines 339-340: Why is the < 1 kDa peptides content of F higher than that of A+F, please explain it.

The < 1 kDa peptides content of F is not higher than A+F. As reported in the text and in the Figure 2, < 1 kDa peptides content of F is 72 % while that of A+F is 76 %.

Reviewer 2 Report

The authors have studied the preparation, characterization and in vitro stability of a novel 2 ACE-inhibitory peptide from soybean protein. Although several ACE-inhibitory peptides have been identified in soybean protein hydrolysates, their exact mechanism of action was not understood. Here, the authors have used commercially available food-grade proteases for the preparation of soybean hydrolysates with good ACE inhibitory activity. An ACE-inhibitory peptide was identified and its mechanism of action was established by enzyme kinetic study, and also determined its stability toward simulated gastrointestinal digestion. The study is novel and the research is within the scope of the Journal. So my recommendation is that, this paper should be accepted for publication in Foods. However, the manuscript needs some minor modifications/clarification, which needs to be taken care off and the paper needs to be modified accordingly.

Following are my specific questions/and general comments;

1.      What is the rationale behind chemically synthesising the peptide, when it is generated in huge quantities from soybean?

2.      How the percentage of different peptides (>10 kDa, >5 kDa, >1 kDa and <1 kDa, Figure 2) was determined? It is not mention in the methodology part?

3.      In Figure 4, the interpretation of MS/MS data needs to be checked thoroughly. For example, the b-series and y-series ions are wrongly assigned.  As per the assignment in the figure, y3 ion should not be less than y2 ions. So, please check and rectify?

4.      The enzyme activity at different pH (8.5 Vs 7.5), time duration (8 hrs Vs 24 hrs), and incubation temperature (55 Vs 50 DegC), may not be the reason for producing different peptides from the same enzymes. This will not produce the peptides having only competitive or non-competitive inhibitions. Please revise the concluding part. 

5.      Values in text (see line nos. 288 and 298) are different then in the Table 1. Please check and rectify.

6.      ‘The maximal ACE inhibition of NDRP digested sample was found to be 82.47 ± 1.07 % corresponding to 93 % of its starting activity….’ It is not clear, how 82.47 ± 1.07 % is corresponding to 93 % of its initial activity? Correspondingly, revise the conclusion part. 

7. What is the result of this supplementary Figure S10? It is HPLC of 8 cycle of Edman degradation, but what is the result? Please explain in text. 

8. Check for sentence structure, for example line nos. 399-400.

9.  Units should be same throughout the text.  600L/h can be written as 600 L h-1 (line No. 195).

Author Response

Referee 2

We thank the Referee for carefully reading our manuscript and for giving such constructive suggestions.

  1. What is the rationale behind chemically synthesising the peptide, when it is generated in huge quantities from soybean?

Peptides were synthesized in small amounts by solid phase synthesis for two reasons:

to prove unequivocally their structures;

to obtain quickly a sufficient amount of highly pure peptides to perform biological assay. Isolation of peptides from soy hydrolysates, a very complex matrix, is not trivial involving several steps of separation and purification. It is much more tedious and time consuming than chemical synthesis.

  1. How the percentage of different peptides (>10 kDa, >5 kDa, >1 kDa and <1 kDa, Figure 2) was determined? It is not mention in the methodology part?

The percentage of different peptides (>10 kDa, >5 kDa, >1 kDa and <1 kDa) were evaluated as weight percentage (w/w %) with respect to the weights of the starting hydrolysates. This experimental detail has been added in the experimental part.

  1. In Figure 4, the interpretation of MS/MS data needs to be checked thoroughly. For example, the b-series and y-series ions are wrongly assigned. As per the assignment in the figure, y3 ion should not be less than y2 ions. So, please check and rectify?

We apologize for this mistake. MS/MS data have been carefully checked and assignments of the b-series and y-series ions have been revised. In particular, we have corrected the labelling of y2 and y3 which were inverted.

  1. The enzyme activity at different pH (8.5 Vs 7.5), time duration (8 hrs Vs 24 hrs), and incubation temperature (55 Vs 50 DegC), may not be the reason for producing different peptides from the same enzymes. This will not produce the peptides having only competitive or non-competitive inhibitions. Please revise the concluding part.

We agree with the Referee. The differences between our work and that of Xu et. al. include not only the hydrolytic conditions but also the starting material, the separation techniques (ultrafiltration membranes with different molecular weight cut-off i.e. 3 kDa vs 1 kDa) and the chromatographic methods used. All these parameters could have influenced the release and/or the identification of specific peptide sequences acting with different mechanisms with respect to the ones that we have found. We have revised the text accordingly.

  1. Values in text (see line nos. 288 and 298) are different then in the Table 1. Please check and rectify.

We apologize for the mistake that was emended in the revised version.

  1. ‘The maximal ACE inhibition of NDRP digested sample was found to be 82.47 ± 1.07 % corresponding to 93 % of its starting activity….’ It is not clear, how 82.47 ± 1.07 % is corresponding to 93 % of its initial activity? Correspondingly, revise the conclusion part.

We compared the ACE inhibitory activity of a pure sample of NDRP with that of a sample of NDRP submitted to the in vitro simulated gastrointestinal digestion. The latter activity resulted to be ca. 94% of the former, indicating that the peptide maintains most of its bioactivity and it is stable towards digestion. According to Reviewer suggestion, we have better explained this result in the text.

  1. What is the result of this supplementary Figure S10? It is HPLC of 8 cycle of Edman degradation, but what is the result? Please explain in text.

In the Experimental Section we have added further details on the Edman degradation as well as on the results obtained from the HPLC analysis reported in the Supplementary Material.

  1. Check for sentence structure, for example line nos. 399-400.

The sentence has been checked and changed.

  1. Units should be same throughout the text. 600L/h can be written as 600 L h-1 (line No. 195).

In the revised manuscript, 600 L/h has been changed in 600 L h-1. We have checked the units throughout the text.

Reviewer 3 Report

The manuscript is about identifying ACE-inhibitors peptides in soy proteins hydrolysed by three enzymatic processes. The study was well conducted, using successive steps to fractionate and identify  ACE inhibitory peptides. The stability of the peptides to digestion was also evaluated.

Some remarks about the manuscript: 

- The electrophoresis is very poor and should be improved, as well as its discussion.

- Table 1 is unnecessary since all the results are n the text. 

- Lines 334- 338: This paragraph is very confusing. No emulsifying results exist, and foaming and emulsifying properties do not explain or relate to intensive hydrolysis. 

- There are excessive details in the supplementary material

Author Response

Referee 3

We thank the Referee for his/her positive comments. We have greatly appreciated his/her help in revising this manuscript

- The electrophoresis is very poor and should be improved, as well as its discussion.

We are aware that electrophoresis is poor (bands blurry and dull). This frequently occurs when low molecular weight peptides, such as those contained in our soy hydrolysates, are detected. Unfortunately, any attempt to improve it was unsuccessful. However, we think that this result highlights that in the reported experimental conditions both Alcalase® and Flavourzyme®, as well as their combination, led to the complete hydrolysis of the main proteins contained in soy. This finding may be relevant as some of these proteins are considered allergenic. A further characterization of the molecular weight distribution of the hydrolysates was then carried out by means of the ultrafiltration technique. As suggested, we have improved the discussion in the main text.

- Table 1 is unnecessary since all the results are n the text.

Table 1 cannot be removed to meet the request of Referee 1 about statistical analysis of data of Tables 1 and 2.

- Lines 334- 338: This paragraph is very confusing. No emulsifying results exist, and foaming and emulsifying properties do not explain or relate to intensive hydrolysis.

See response to Reviewer n. 1, remark n. 6.

The sentence referring to the emulsifying and foaming properties of the hydrolysates has been erroneously inserted in the text. It has been deleted in the revised manuscript. We apologize for the mistake due to the fact that our hydrolysates seem to possess functional properties (evaluated as emulsifying and foaming capacity) suitable for a possible scale-up. However, these issues have not been further deepened because outside the scope of our research.

- There are excessive details in the supplementary material.

Excessive details in the Supplementary Material section have been cut out. In particular, data regarding Foaming Properties, Degree of Hydrolysis, Amino Acid Analysis and Peptide Loading have been removed.